# Cost-Effective Production of TiO_2_ with 90-Fold Enhanced Photocatalytic Activity Via Facile Sequential Calcination and Ball Milling Post-Treatment Strategy

**DOI:** 10.3390/ma13225072

**Published:** 2020-11-10

**Authors:** Anantha-Iyengar Gopalan, Jun-Cheol Lee, Gopalan Saianand, Kwang-Pill Lee, Woo-Young Chun, Yao-long Hou, Venkatramanan Kannan, Sung-Sik Park, Wha-Jung Kim

**Affiliations:** 1School of Architecture and Civil Engineering, Daegyeong Regional Infrastructure Technology Development Center, Kyungpook National University, Daegu 41566, Korea; algopal99@gmail.com (A.-I.G.); kplee@knu.ac.kr (K.-P.L.); 2Department of Architecture, Seowon University, Cheongju 28674, Korea; uggenius@hanmail.net; 3Global Centre for Environmental Remediation (GCER), Faculty of Science, The University of Newcastle, Callaghan 2308, NSW, Australia; SaiAnand.Gopalan@newcastle.edu.au; 4Real-scale Fire Testing; Research Center, Korea Conformity Laboratories, 33-72, Eonjang 1-Gil, Samcheok-si, Gangwon-do 25913, Korea; cwy@kcl.re.kr; 5Department of Civil Engineering, Kyungpook National University, 80 Daehakro, Buk-gu, Daegu 41566, Korea; hylmm8988@hotmail.com (Y.-l.H.); sungpark@knu.ac.kr (S.-S.P.); 6Department of Physics, SCSVMV Deemed University, Kanchipuram 631561, India; kv@kanchiuniv.ac.in

**Keywords:** titanium dioxide, low-cost, calcination, ball milling, post-treatment

## Abstract

Titanium dioxide (TiO_2_), the golden standard among the photocatalysts, exhibits a varying level of photocatalytic activities (PCA) amongst the synthetically prepared and commercially available products. For commercial applications, superior photoactivity and cost-effectiveness are the two main factors to be reckoned with. This study presents the development of simple, cost-effective post-treatment processes for a less costly TiO_2_ to significantly enhance the PCA to the level of expensive commercial TiO_2_ having demonstrated superior photoactivities. We have utilized sequential calcination and ball milling (BM) post-treatment processes on a less-costlier KA100 TiO_2_ and demonstrated multi-fold (nearly 90 times) enhancement in PCA. The post-treated KA100 samples along with reference commercial samples (P25, NP400, and ST01) were well-characterized by appropriate instrumentation and evaluated for the PCA considering acetaldehyde photodegradation as the model reaction. Lattice parameters, phase composition, crystallite size, surface functionalities, titanium, and oxygen electronic environments were evaluated. Among post-treated KA100, the sample that is subjected to sequential 700 °C calcination and BM (KA7-BM) processes exhibited 90-fold PCA enhancement over pristine KA100 and the PCA-like commercial NP400 (pure anatase-based TiO_2_). Based on our results, we attribute the superior PCA for KA7-BM due to the smaller crystallite size, the co-existence of mixed anatase-srilankite-rutile phases, and the consequent multiphase heterojunction formation, higher surface area, lattice disorder/strain generation, and surface oxygen environment. The present work demonstrates a feasible potential for the developed post-treatment strategy towards commercial prospects.

## 1. Introduction

Titanium dioxide (TiO_2_) photocatalysis is predominantly being utilized as a pivotal solution to mitigate issues related to energy scarcity and environmental protection [1,2,3,4,5,6,7]. The increasing photocatalytic application prospects of TiO_2_ create demand for its mass production and thus a large number of companies produce varieties of TiO_2_ products [8,9,10]. However, there exist wide variations in the photocatalytic activities (PCA) as well as the cost per kilogram (Kg) among the commercial products [11]. The photocatalytic efficiency of various TiO_2_ is compared with relevance to a particular model photoreaction, for example, gaseous acetaldehyde (AD) degradation or dye degradation in solution, etc. [12]. One should keep in mind that PCA of TiO_2_ depends on a large number of factors that include crystalline phase structure, phase composition, particle size, surface area, porosity, bandgap, morphology, and surface-bound hydroxyl species [12,13]. These factors vary among the commercial TiO_2_ products leading to differences in the PCA. There is a controversy in the literature on deciding the combination of the above-mentioned factors that determine the PCA of the particular commercial TiO_2_ [14,15]. Among the commercial TiO_2_ products, Degussa P25 TiO_2_ has been mostly considered as the research standard because it has substantially higher PCA than most of the other available samples of TiO_2_ [16]. For the photocatalytic application purposes and also mitigating the inherent problems associated with pristine TiO_2_, post-treatment processes of commercial TiO_2_ have been utilized to enhance the PCA of TiO_2_. The associated total costs (the price of the TiO_2_ and the post-treatment) will ultimately widen the potential utilities in real-life applications (Scheme 1). Hence, it is challenging to design a simple and cost-effective post-treatment strategy towards enhancing PCA of a commercially low-cost TiO_2_ to the level of P25 TiO_2_ or relatively inexpensive TiO_2_.

Considerable research activities are being focused on the enhancement of photocatalytic properties of TiO_2_ that mostly involve pre-synthetic or in-situ modification strategies [17]. Reports on post–modification properties of TiO_2_ are scarce, especially with relevance to modification of commercial TiO_2_. One must note that the smaller particle size of TiO_2_ can result in larger surface area and higher photocatalytic activity. Towards this direction, the grinding process of the powder, like dry/wet ball milling (BM), was employed to obtain smaller particle size and narrower size distribution [18,19]. On the other hand, it is to be noted that phase, crystallite size, and crystallinity of TiO_2_ play an important role in the enhancement of photocatalytic activity, and these modifications were attempted by the calcination process. The effect of calcination temperature on a phase transition of TiO_2_ from anatase to rutile was studied in temperatures ranging from 350 to 750 °C [20]. A phase mixture of anatase and brookite was found at a calcination temperature of 200 °C, a mixture of three phases (anatase, rutile, and brookite) was found at 600 °C, and a complete transformation into rutile phase was noticed at 800 °C [21]. The increase in crystallite size and crystallinity of TiO_2_ was identified by increasing the calcination temperature. However, an explanation of the effect of the changes in crystallite size and phase transformation of TiO_2_ on the photocatalytic activity has not been detailed or correlated. Recently, a hydrothermal post-treatment was adopted to enhance the photocatalytic activity of TiO_2_ [22]. Taking photo-decolorization of Rhodamine B dye as the model reaction, a nearly 10 times increase in photoactivity was reported for the hydrothermal-treated TiO_2_ with water as solvent as compared with that of the untreated ones. The enhancement in photoactivity was ascribed due to the formation of desirable anatase phase, porous structure, large surface area, small particle size, and hydrophilic surface. Literature reveals that new and cost-effective post-treatment of TiO_2_ is highly warranted, especially on commercial products, towards realizing significant enhancement in the photocatalytic activities. In this work, we have chosen KA100 TiO_2_ with a price of 4$/Kg and post-treated with a combination of calcination and BM processes to enhance the PCA of KA100 to the level closer to commercially available expensive TiO_2_ samples such as P25 (~54$/Kg), ST01 (~75$/Kg), and NP400 (~54$/Kg). The post-treatment conditions for KA100 which generate enhanced PCA with P25 or other listed commercial samples have been optimized and the reasons for the enhanced PCA are explained based on the modification of properties derived through post-treatment and the enhancement in the PCA for acetaldehyde (AD) degradation.

## 2. Materials and Methods

### 2.1. Chemicals

The commercial TiO_2_ samples, ST01 (Ishihara Sangyo, Yokkaichi, Japan), P25 (Degussa, Frankfurt, Germany), NP400 (Bentech Frontier, Yongbong-ro Buk-gu, Korea), and KA100 (Cosmo chemicals, Seoul, Korea), were purchased and used.

### 2.2. Calcination and Ball Milling

KA100 was calcined at different temperatures and the samples were designated based on the temperature of calcination. The calcined sample was subjected to wet BM using a ball milling machine (IBMT-38-4, As One, Osaka, Japan). A typical procedure is outlined. The alumina milling jar (having 1000 mL volume) was filled with 40 g of a calcined sample, 40 g of distilled water, and alumina balls (having a diameter of 10 mm). The rotational speed of the motor in the BM machine was 650 rpm and the BM duration was 3 h. After three hours of BM, the sample was washed with distilled water and dried at 100 °C for 2 h.

### 2.3. Characterization

The crystal structure, phase composition, and crystallinity of the samples were examined by an X-ray diffraction (XRD) analyzer (D/Max-2500, Rigaku, Tokyo, Japan) [23]. The scanning angle 2θ was varied from 5° to 80° with a step of 0.02°. The operating voltage, applied current, and Cu Kα radiation were 45 kV, 40 mA, and 1.5418 Å, respectively. For crystallinity determination, CaF_2_ was used as the standard and the procedure is relayed elsewhere [24,25]. X-ray photoelectron spectrum (XPS) was recorded with an X-ray Photoelectron Spectroscopy (NEXSA, ThermoFisher, Waltham, MA, USA) using an Al-Kα monochromator source (1486.6 eV) [26,27,28]. Raman spectra were recorded in the range 100 to 2700 cm^−1^ using a Raman spectrometer (inVia reflex, Renishaw, Wotton-under-Edge, UK), equipped with a 536 nm laser [29]. The band gaps of the samples were determined by Ultraviolet-visible diffuse reflectance spectroscopy (S-3100, Seoul, Scinco, Korea) in the wavelength region 200 to 700 nm.

### 2.4. Photocatalytic Decomposition of Acetaldehyde (AD)

The photocatalytic AD degradation experiment was designed and carried out by the gas bag A method standardized by the Korea Photocatalyst Association. Typically, about 1.0 g of the photocatalyst was dispersed into 0.5 g of water to obtain a stable slurry. The photocatalytic slurry was applied on the surface of a pre-cleaned circular glass plate (90 mm of diameter) and dried at 100 °C for 2 h. The Tedlar bag (3 L volume) with the coated glass plate was placed in a stainless-steel box. The diluted AD gas with air having an initial AD concentration of 100 ppm was inducted within the Tedlar bag set-up. The sample was irradiated with a UV lamp (Sankyo Denki, Kanagawa, Japan, UV wavelength from 310 to 400 nm, 15 W) fixed in the stainless-steel box. The ultraviolet intensity of the lamp was 1.0 mW/cm^2^ on the sample surface. The concentration of AD gas was monitored with the gas analyzer (Gas sampling pump kit, GV-100 s, Gastec Corporation, Kanagawa, Japan) every 30 min over 2 h.

## 3. Results and Discussion

### 3.1. Influence of Calcination and BM Processes on Microstructural and Electronic Properties

Figure 1 presents the XRD patterns of pristine KA100, calcined KA100 at different temperatures, few calcined + BM, and reference TiO_2_ (P25, NP400, and ST01) samples. XRD patterns were consistent with data of the TiO_2_ comprising of a mixture of anatase (A) (major weight proportion, ~97%) and rutile (R) (minor weight proportion, ~3%) (Table 1). Particularly, the KA7-BM showed a higher R phase proportion (3.30%). To note, the R% increased from 2.78% to 3.30% when the KA7 was subsequently subjected to BM. It is inferred that the mechanical force/energy involved in BM could transform A to R phase. It must be noted that NP400 and ST01 samples were pure A samples, whilst P25 comprises of 77.90% A and 22.10% R phases. Generally, TiO_2_ exhibits higher photocatalytic activities because of the synergistic effects from the two phases, band alignments, and the probable heterojunction/interfacial electron transfer [10,23,30]. TiO_2_ with a mixture of A and R compositions has been reported to exhibit lower rates for the recombination of photo-generated electrons and holes, and R contributes significantly to better light absorption [10]. The literature suggests that a mass ratio of 3:7 (R to A) is the best for achieving high photocatalytic activities [31]. However, other reports detail that even a lesser proportion of R can also yield improved photoactivities, and the parameters like crystallite size, surface area, and optical absorption will determine the PCA in such cases [18]. Also, a mixture of A and brookite (B), as well as R and B, can result in fast photodegradation rates of organic molecules [32]. In the present case, non-calcined, calcined, and calcined + ball milled KA100 samples did not show even a trace of the B phase. 

Most commonly, the particle size is determined using X-ray diffraction measurements and by the help of the Scherrer equation [33]. However, keeping note that Scherrer equation is related to a sharp peak of X-ray diffraction and there are limitations based on the use of non-monochromatic X-ray source, dependence on the instrument as well as the relationship between signal, sample to criterion noise, and influences by peak broadening, considerations are given to improve the reliability of size determinations by other methods [34]. Modification through the Williamson-Hall (W-H) method was developed by considering the contributions of additive effects of different angular dependencies of the size and strain/disorder functions for reflections [35]. To note, the W-H method for crystallite size and strain analysis is still underutilized as compared to the Scherrer method and this may be because there are several limitations of the W-H approach and few further modifications of the W-H approach were suggested [36]. The Rietveld method (R-M) has been introduced considering a whole-pattern fitting approach instead of single-peak analysis [37]. The main advantage of the R-M is that it can effectively minimize or eliminate the inaccuracies arising from preferred orientation, particle statistics, micro-absorption, peaks overlapping, and detection of amorphous phase and trace phases [38]. Based on the R-M and refinements, the amorphous and the nanocrystalline alloys such as Fe–B–Nb alloy structure [39] and Ni-Mo alloys [40] were analyzed [41]. However, one can notice at least two important preconditions for quantitative crystallite size analysis such as: (1) the quantified phase is the crystalline phase, and (2) the crystal structure is known [42]. Particularly, if amorphous material is involved, quantitative results could not be achieved directly and with precision. Keeping all these advancements happening for the determination of crystallite size, one can notice through the literature that the Scherrer equation is almost universally used for the determination of crystallite size, of course, despite its limitations, and reliability of meaningful size information derived from such calculations. In this work, it is reasonable to consider the formation of the amorphous state of any of the TiO_2_ phases (anatase, rutile, brookite, or any other) upon post-treatment of TiO_2_ samples. Hence, the Scherrer method was used in this work for obtaining a comparative analysis of the crystal structural parameters among the samples.

The crystallite size was calculated using the Scherrer formula which suggested that the crystallite size of KA100 increases marginally from 60 nm (for non-calcined) to 65 nm (for the calcinations temperature of 1000 °C) with calcination treatment. The KA7-BM sample had the smallest size amongst the investigated KA100 samples. The decrease in particle size after BM has been reported and the extent of decrease depends on the energy of the BM [20]. A drastic increase in PCA of close to one hundred times that of the untreated sample was witnessed for the BM processed samples. The crystallite size of the ST01, NP400, and P25 is much smaller (6, 17, 20 nm, respectively) than the calcined KA100 and BM processed samples. The lattice parameters of crystalline KA100 did not change either by calcination or by B, signifying that these post-treatments did not influence the crystal structure. The A crystallinity was estimated by the full width at half maximum (FWHM) at the 101 main peaks with reference to CaF_2_ as the standard (Table 1 and Table 2). Whilst the KA4 is 100% crystalline, the value decreased to 72.0% for KA7. However, the crystallinity increased to 89.2% upon subsequent BM. The amorphous and metastable srilankite (S) (TiO_2_-II phase, orthorhombic) formation could be possible upon calcination at 700 °C [32,43]. This could be the reason for the decreased crystallinity at 700 °C. However, subsequent BM (mechanical activation) could induce transformations of amorphous S to crystalline S and R phase formation. The increased crystallinity and R proportion for KA7-BM are evident from Table 1 and Table 2. The decrease in particle size upon BM of KA7 could be correlated to the phase transformations (A to S to R) as detailed above [44]. It is therefore envisaged that subsequent BM processing of KA7 caused the deconstruction and then reconstruction of the crystal grains [45]. Thus, KA7-BM is expected to possess a multiphase microstructure [43]. Specific surface area and pore size of the selected samples were obtained from the N_2_ adsorption-desorption isotherms and presented (Table 1 and Table 2). KA7-BM showed a nearly 10% increase in surface area as compared to KA7 (Table 1 and Table 2). The larger surface area of the KA7-BM is correlated to the smaller crystallite size as compared to KA7.

The optical bandgap of different samples was investigated by diffuse reflectance measurements and the derived Tauc plots (Figure 2). It was noticed that the KA100 samples comprising of A-R phases have an energy bandgap between 3.06 and 3.03 eV. Keeping in mind that the bandgap of a semiconductor can be influenced by several factors such as doping, synthetic parameters, lattice mismatch [46], crystallite size [47], annealing/calcination temperature [48], and phase composition, and also based on the recent reports that inform “the bandgap determination through extrapolation of Tauc segment to the hv (X) axis is not appropriate for samples containing more than one prominent optical absorbing center/or phase” [49,50], the bandgap was also determined the by extrapolation of the linear part of the dependence to the baseline (Figure 2). The optical band as determined by the Tauc plot and baseline extrapolation method is referred to as BG-T and BG-BE, respectively. Apparently, the bandgap was not altered much by the post-treatment processes for KA100 (Table 1 and Table 2).

The peak positions of Raman bands corresponding to A phase in non-calcined KA100 are compared to other post-treated KA100 samples (Figure 3, Table 3). The Raman bands at 142.02 cm^−1^ (Eg), 195.5 cm^−1^ (Eg), 394.6 cm^−1^ (B1g), 516.3 cm^−1^ (A1g + B1g), and 638.5 cm^−1^ (Eg) correspond to the Raman allowed modes of A TiO_2_. However, the Raman band at 105 cm^−1^, which is a unique feature of R-TiO_2_, is not observed or it has very minimum intensity. The Raman B1g mode showed red (lower wavenumber) shifting upon a post (heat)-treatment, whilst A1g + B1g and Eg bands showed blue (higher wavenumber) shifts (Figure 3). Factors such as crystallite size, the composition of mixed phases (A to R), the nature and magnitude of the defects in the crystal structure, and stoichiometry deviation can influence these Raman bands either alone or in conjunction. Raman spectrum of post-treated KA100 samples informs that post-treatment of KA100 could have caused one or more of the above-mentioned modifications. 

The Ti 2p XPS spectra of KA4, KA7, and KA8 samples exhibited Ti 2p3/2 and Ti 2p1/2 doublets and a satellite peak, which are characteristic peaks associated with the Ti4+ valence state on lattice oxygen of TiO_2_ [51] (Table 2 and Figure 4). The appearance of Ti 2p3/2 and Ti 2p1/2 peaks (with a separation close to the standard value of 5.6 to 5.7 eV), along with a well-known charge transfer shake-up satellite peak (~13.2 eV higher than the main peak) are characteristics of TiO_2_ (Table 2). The Ti2p XPS spectrum of KA7-BM shows additional shoulders around 463 and 457 eV that could be assigned to the presence of Ti^3+^ valence state and associated with oxygen vacancy as the defect in TiO_2_ crystal lattice [52]. Besides, the Ti2p XPS spectrum of KA7 + BM shows the satellite/shake-up peak at a larger distance (13.50 eV) from the main Ti 2p peak (Table 2). Because the position of the shake-up satellite peak is correlated to the excitation of a valence electron to a previously unoccupied state by the outgoing Ti 2p photoelectron, it is envisaged that the charge transfer characteristics of KA7 + BM can be different from the other samples, possibly due to the lattice disorder and the subtle deviations in the electronic structure. The calcined KA100 samples exhibit a O1s peak at 529.78 eV (Figure 4) that is assigned to oxygen bound to tetravalent Ti ions. The observed difference in the O1s peak position KA7 + BM (529.68 eV) correlates with the probable lattice disorder as inferred from the Ti 2p electronic environments (Figure 3). Besides, the presence of a shoulder at ~533 eV suggests that the BM processed TiO_2_ surface is partially covered with hydroxide OH groups [53].

### 3.2. Photocatalytic Degradation of AD

Figure 5a–c depicts the changes in the AD concentration over time due to the photodegradation by the various TiO_2_ samples under investigation. Typically, non-calcined KA100 decomposes 50% of AD after 2 h, whilst 100% of degradation was evident for KA100 at a similar period (Figure 4a). Interestingly, KA7 + BM degrades AD at a faster rate, and 100% AD decomposition was witnessed by 30 min ahead of KA7 (Figure 5b). P25 and ST01 exhibited much faster AD photodegradation, requiring nearly 60 min for 100% decomposition. However, the PCA of KA7 + BM is similar to NP400 (Figure 5a–c). More quantitative evaluation and comparison of PCA of the studied samples were done by deriving the first-order rate constant of the AD photodegradation (kAD). The values of kAD were determined from the slope of the plots, logarithmic concentration of AD at a time (lnC) versus time (Figure 6a–c). Figure 5d presents the kAD of the non-calcined and calcined samples. The KA7 + BM showed the highest kAD among the studied samples. Figure 5e compares the kAD values between KA7, KA7 + BM, and pristine commercial TiO_2_ samples (P25, ST01, and NP400). One must note that KA7 has slightly higher kAD than KA8. Due to the higher kAD of KA7 and lower calcination temperature requirement, KA7 was considered for further BM treatment. Hence, the comparison chart included only the kAD values of KA7 and KA7 + BM. The comparison reveals that the PCA of KA7 is increased ~60 times as compared to KA100 and a ~90 times increase was witnessed by the subsequent BM process. The simple BM process of KA7 caused a 45% increase in the kAD of KA7. Interestingly, the photocatalytic performance of KA100 (less expensive, 4$/Kg) could be improved like NP400 (more costly, 54$/kg) by simple post-treatment processes (calcination at 700 °C and subsequent BM). Thus, nearly 13 times less expensive KA100 could be transformed to show the PCA similar to NP400, and nearly 67% enhancement as that of PCA of much costlier ST01 and P25. We strongly believe that further tuning the BM parameters such as power (the energy involved), time, etc., of the KA100 post-treatment process could result in PCA similar to P25 and STO1. On perusal of Figure 5e, it is evident that KA7-BM exhibited the superior PCA amongst the post-treated KA100 samples. Literature suggests that improved photoactivity for TiO_2_ could arise through various factors such as mixed-phase existence, high surface area, high crystallinity, presence of surface defects, and hydrophobic–hydrophilic transformation [54,55]. A comparative photocatalytic activity study has been performed earlier between two commercial TiO_2_ powders (Cristal Global, Millennium PC500, and Evonik, P25), taking AD degradation in the gas phase as the model photoreaction [12]. It has been notified that the quantum efficiency of P25 is superior to PC500 based on the higher available surface area. However, the better PCA for PC500 as compared to P25 is attributed to the presence of surface OH groups and alternative reaction pathways even with a lower amount of photogenerated holes. In the present work, the superior photocatalytic performance of KA7-BM (90-fold enhancement over KA100) is attributed to synergistic factors that operate because of the smaller crystallite size, co-existence of A-S-R phases, and the consequent multiphase heterojunction formation, higher surface area, lattice disorder/strain generation, and surface oxygen environment. The outcome of the research paves the way for the development of a simple post-treatment strategy to enhance photocatalytic performance on a cost-effective basis and the consequent commercial prospects for large-scale production of cheaper TiO_2_ with enhanced photocatalytic performances.

## 4. Conclusions

In summary, we have successfully prepared a facile and low-cost version of TiO_2_ through a unique post-treatment strategy involving sequential calcination and ball milling pulverization. The influence of calcined, non-calcined, and calcined and ball-milled samples were analyzed for their microstructural and electronic properties, and the photocatalytic performances were compared with reference commercial samples of TiO_2_. The experimental results suggest that the as-developed cost-effective TiO_2_ sample (KA7) possessed excellent photocatalytic activities (90-fold enhancement) for the degradation of acetaldehyde compared to pristine KA100 and were less expensive than their commercial counterparts (P25, NP400, and ST01).

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
