# Peer review of "Cost-Effective Production of TiO2 with 90-Fold Enhanced Photocatalytic Activity Via Facile Sequential Calcination and Ball Milling Post-Treatment Strategy"

_materials, 2020, doi:10.3390/ma13225072_

Round 1
Reviewer 1 Report
The article is basically written correctly and contains interesting results that may have application significance.
However, there are formal errors in the text.
Moreover, the structural analysis needs to be significantly improved.
1 - page 3, line 87 - "dwell time" - usually "time per step"
The time of 1.5 seconds results in a very poor recording of the diffraction pattern. The diffraction patterns described in the article are therefore very weak and poorly defined.
2 - page 3, line 88 - "Cu Kα radiation were 40 kV, 200 mA, and 1.5406 Å, respectively."
the current is probably 20 mA
The wavelength for the copper anode, without the monochromator used, is normally given a weighted average of 1.5418 Å.
3 - page 4, Fig. 1 - schould be improved (Quality and description)
4 - page 4, line 128 - "crystallite size was calculated using the Scherrer formula"
The analysis of the crystallite size, performed only with the use of the Scherrer formula, is incorrect and burdened with a very large error. The size of the crystallites is also influenced by lattice strains which essentially affect the obtained results. There are many studies in the literature that describe calculation methods and present a comparative analysis of the results obtained with various assumptions and parameters. For the correct calculation of the crystallite size, it is necessary to use a structural standard, eg LaB6, Si etc, which we use to eliminate the widening of diffraction peaks caused by instrumental factors.
In this context, specifying the crystallite size with an accuracy of 0.01 nm is a misunderstanding!
Sample literature is .....
M. Karolus, E. Łągiewka, Crystallite size and lattice strain in nanocrystalline Ni-Mo alloys studied by Rietveld Refinement, J. Alloys Compd. 367 (2004) 235–238. https://doi.org/10.1016/j.jallcom.2003.08.044.
Williamson GK, Hall WH. X-ray line broadening from filed aluminium and wolfram. Acta Metall. 1953;1(1): 22–31.
5 - page 5, table 1
The results presented in Table 1 do not they look believable.
Crystallite size values (accuracy of their determination), lattice constants (identical ??? how were they calculated ?? there is no description of the methodology), quantitative analysis (accurasy) raise doubts. When presented as a weak diffraction patterns, they presented the results are not reliable.
Reviewer 2 Report
The authors investigate several post-treatment processes for a commercial low-cost TiO2 sample to significantly enhance the PCA to the level of expensive commercial TiO2. The post-treatment strategy included calcination and ball milling pulverization. The total enhancement in the PCA was obtained as high as 90 times. The post-treated TiO2 samples and referenced high-cost commercial samples were studied by XRD, BET, DRS, XPS, and Raman analysis and evaluated for the PCA. The best post-treatment conditions were found based on the smaller crystallite size, coexistence of three phases, and the consequent multiphase heterojunction formation, higher surface area, lattice disorder/strain generation, and surface oxygen environment.
The manuscript is well-written and organized. The authors reveal a superior post-treatment process and make the comprehensive study of the characteristics. There are a few points which can be improved in this manuscript. First, there are a number of other effective post-treatment processes, like the one reported in New J. Chem., 2020, 44, 1942, and many others. The manuscript could be improved by additional references and a comparison of post-treatment processes for the photocatalytic activity of TiO2 in Introduction. Second, it is clear from Figure 5d that the sample with the 800oC calcination has very close values to the one with 700oC but it is not shown in Figure 5e. It is not clear from the text how the ball milling changed the rate constant for this sample. After these minor improvements, the manuscript can be published in Materials.
Reviewer 3 Report
The proposed manuscript describes the simple strategy for improving photocatalytic performance of KA100 TiO2 with a price of 4 $/Kg to the level closer to much more expensive commercially available TiO2 samples.
The paper is well-written, the results are discussed in details. The conclusions are strongly supported by the obtained results.
Therefore, it is my pleasure to recommend it for publication in Materials after adressing the following issues:
- since Eg determination from the Tauc plots may sometimes lead to erroneous estimates (especially when the absorption at lower energies is meaninful) it would be beneficial to verify, if the strategy proposed by Macyk et al. (https://doi.org/10.1021/acs.jpclett.8b02892) suggesting the extrapolation of the linear part of the dependence to the baseline rather than to the energy axis can be useful in the present case
- since the first-order kinetics was assumed for the determination of rate constatnt of AD photodectradation process, logaritmic dependence between C and t would be beneficial
Round 2
Reviewer 1 Report
The corrections and additions introduced in the text are acceptable.